# Dimerization and auto-processing induce caspase-11 protease activation within the non-canonical inflammasome

Connie Ross, Amy H Chan, Jessica Von Pein, Dave Boucher*, Kate Schroder* (ORCID)

Caspase-11 is a cytosolic sensor and protease that drives innate immune responses to the bacterial cell wall component, LPS. Caspase-11 provides defence against cytosolic Gram-negative bacteria; however, excessive caspase-11 responses contribute to murine endotoxic shock. Upon sensing LPS, caspase-11 assembles a higher order structure called the non-canonical inflammasome that enables the activation of caspase-11 protease function, leading to gasdermin D cleavage and cell death. The mechanism by which caspase-11 acquires protease function is, however, poorly defined. Here, we show that caspase-11 dimerization is necessary and sufficient for eliciting basal caspase-11 protease function, such as the ability to auto-cleave. We further show that during non-canonical inflammasome signalling, caspase-11 self-cleaves at site (D285) within the linker connecting the large and small enzymatic subunits. Self-cleavage at the D285 site is required to generate the fully active caspase-11 protease (proposed here to be p32/p10) that mediates gasdermin D cleavage, macrophage death, and NLRP3-dependent IL-1$\beta$ production. This study provides a detailed molecular mechanism by which LPS induces caspase-11–driven inflammation and cell death to provide host defence against cytosolic bacterial infection.

## Introduction

Caspase-11 is a key mediator of the murine innate immune response to cytosolic Gram-negative bacterial pathogens, by allowing the recognition of bacterial LPS (Hagar et al, 2013; Kayagaki et al, 2013; Shi et al, 2014). Cytosolic LPS engages the non-canonical pathway of inflammasome activation, whereby active caspase-11 cleaves its substrate gasdermin D (GSDMD) to generate a GSDMD-p30 fragment that forms pores in the plasma membrane (He et al,

2015; Kayagaki et al, 2015; Shi et al, 2015; Aglietti et al, 2016; Ding et al, 2016; Liu et al, 2016). In many cells, GSDMD pores trigger an inflammatory form of cell lysis termed pyroptosis (He et al, 2015; Kayagaki et al, 2015; Shi et al, 2015; Aglietti et al, 2016; Ding et al, 2016; Liu et al, 2016), whereas in neutrophils, these pores allow the extrusion of neutrophil extracellular traps (Chen et al, 2018a). GSDMD pores also indirectly activate the NLRP3 inflammasome to generate active caspase-1 (Ruhl & Broz, 2015), which cleaves pro-IL-1$\beta$ to its mature form that is secreted (Schroder & Tschopp, 2010; Monteleone et al, 2018). Although the signalling pathways up- and downstream of non-canonical inflammasome assembly are increasingly well understood, the molecular events required for caspase-11 activation within the non-canonical inflammasome are unclear. LPS is proposed to be a direct ligand for caspase-11, wherein LPS interaction with the caspase-11 CARD domain facilitates activation of the protease domain (Shi et al, 2014), via an undetermined mechanism.

Caspase-11 belongs to the caspase family of cysteine-aspartate proteases that are involved in diverse cell death signalling pathways. Caspase-11 is most closely related to the other murine inflammatory caspase, caspase-1. Caspase-11 functions as a signal initiator; similar to caspase-1, and the apoptotic caspases, caspase-8 and -9. The proteolytic activities of caspase-1, -8, and -9 are governed by two interrelated molecular processes, dimerization and linker processing (Boatright et al, 2003; Broz et al, 2010; Oberst et al, 2010; Boucher et al, 2018). These caspases are initially produced as monomeric zymogens that are recruited to multimeric signalling complexes via their N-terminal domains (Pop et al, 2006; Boucher et al, 2018). Caspase clustering within these structures facilitates dimerization of the enzymatic subunits, leading to acquisition of basal proteolytic function (Pop et al, 2006; Boucher et al, 2018). Autoproteolysis of caspase-1, -8 and -9 can then occur either within the linker that connects the protease to its N-terminal recruitment domain (e.g., the caspase-1 CARD-domain linker [CDL]) or the linker that connects the two catalytic subunits (interdomain linker [IDL]). The impact of caspase-1, -8, and -9 linker processing on protease activity and substrate repertoire varies depending on the

Institute for Molecular Bioscience (IMB), IMB Centre for Inflammation and Disease Research, The University of Queensland, St Lucia, Australia

Correspondence: K.Schroder@imb.uq.edu.au
*Dave Boucher and Kate Schroder are equal senior authors

caspase (Pop et al, 2006; Oberst et al, 2010; Pop et al, 2011; Boucher et al, 2018). For caspase-1 and caspase-8, IDL auto-processing broadens the substrate repertoire of these proteases, whereas auto-cleavage of the caspase-1 CDL, or caspase-9 IDL, promotes enzyme dissociation from the signalling complex and protease deactivation (Malladi et al, 2009; Boucher et al, 2018). Murine caspase-11, and its human orthologues caspase-4 and caspase-5, are activated within the non-canonical inflammasome complex, which has a number of unusual features compared to canonical, caspase-1-activating inflammasomes. First, caspases-4/5/11 directly bind to LPS without requiring a traditional receptor or signalling adaptor. Second, non-canonical inflammasome assembly is proposed to generate caspase-11/4/5 oligomers (Shi et al, 2014), as compared with caspase-1 dimers elicited by canonical inflammasomes in macrophages (Boucher et al, 2018). For these reasons, the mechanisms regulating activation of caspase-11/4/5 protease function within the non-canonical inflammasome are suggested to follow a distinct mechanism from that of caspase-1; for example caspase-11 activation may require the formation of oligomers rather than dimers. The stoichiometry of the caspase-11-LPS complex is not defined and it remains unclear whether the higher order caspase-4/5/11 structures induced by LPS are true oligomers, or represent multiple caspase dimers binding to single LPS molecule or LPS aggregate. Caspase-11 harbours multiple candidate sites for auto-cleavage or processing by other caspases (e.g., caspase-1), but the functional impact of cleavage at these sites is poorly defined. Three forms of caspase-11 can be detected by immunoblot using an antibody which detects the large enzymatic subunit: (i) a full-length (43 kD) form; (ii) a shorter (36 kD) form thought to arise from an alternative start codon (methionine 61) within the CARD domain (Kang et al, 2000) that appears unable to bind LPS (Shi et al, 2014); and (iii) a shorter caspase-11 fragment of unknown nature and function, generated during non-canonical inflammasome signalling in macrophages (Kang et al, 2002; Kayagaki et al, 2011).

This study investigates the molecular basis for caspase-11 activation, which is central to non-canonical inflammasome signalling. We demonstrate that caspase-11 dimerization is sufficient for inducing basal caspase-11 activity, such as the ability to self-cleave. IDL, but not CDL, auto-processing of caspase-11 dimers is then required to generate the fully active protease that can cleave GSDMD and thereby mediate non-canonical inflammasome signalling to provide host defence against cytosolic bacterial infection.

## Results

### Caspase-11 is processed to p32 independently of the NLRP3 inflammasome and caspase-1

Cytosolic LPS in macrophages activates caspase-11, leading to pyroptotic cell death and "non-canonical" NLRP3 signalling (Hagar et al, 2013; Kayagaki et al, 2013; Shi et al, 2014). Caspase-11 activity is associated with generation and cellular release of a ~30 kD caspase-11 fragment (hereafter called p32) that is detected with an

antibody against the large subunit, and is presumed to be a product of caspase-11 self-cleavage at undetermined site(s). Cleavage of caspase-11 within either the CDL or IDL at a number of candidate cleavage sites could generate caspase-11 fragments around this size (Fig 1A). It is unclear whether p32 forms part of an active or inactive caspase-11 species. To examine the timing of caspase-11 cleavage and signalling, we first primed WT versus Casp11-deficient bone marrow macrophages (BMMs) with Pam$_3$CSK$_4$ to upregulate caspase-11 expression, and then transfected the cells with ultra-pure K12 E. coli LPS. Non-canonical inflammasome responses were measured by LDH assay for cell lysis, ELISA for IL-1$\beta$ secretion, and Western blotting for cleaved caspase-11, caspase-1, and IL-1$\beta$ over a 24-h time-course. Consistent with other reports (Kayagaki et al, 2011; Shi et al, 2014), LPS transfection induced cell death and IL-1$\beta$ release in a caspase-11–dependent manner (Fig 1B and C). The kinetics of non-canonical inflammasome signalling was slow, occurring over 6–24 h rather than the more rapid kinetics standard for the canonical NLRP3 inflammasome (Boucher et al, 2018). Primed BMMs were also exposed to the NLRP3-specific inhibitor, MCC950 (Coll et al, 2015) before LPS transfection. NLRP3 inhibition did not affect caspase-11–dependent cell death, but abrogated IL-1$\beta$ secretion (Fig 1B and C). LPS-induced, caspase-11–dependent cell death and IL-1$\beta$ secretion was temporally associated with caspase-11 cleavage (Fig 1B–D). LPS transfection promoted the release of both uncleaved (full length and p36) and cleaved (p32) forms of caspase-11 into the cell culture media, and caspase-11 p32 generation occurred concomitant to non-canonical inflammasome signalling outputs (Fig 1B–D). Caspase-11 cleavage to p32 was not a consequence of NLRP3 signalling; p32 generation was not blocked by MCC950, nor was it suppressed in Casp-1$^{C284A}$ BMMs in which the catalytic cysteine of this protease is mutated to disable caspase-1 activity (Fig 1D and E). These data indicate that non-canonical inflammasome activation, and resultant caspase-11 signalling, temporally coincides with the cleavage of caspase-11 to p32, a cleavage fragment that encompasses the caspase-11 large enzymatic subunit. Caspase-11 cleavage to p32 is not mediated by NLRP3, caspase-1, or their downstream pathways, suggesting that p32 may be generated by caspase-11 auto-cleavage during early signalling events within the noncanonical inflammasome.

### Caspase-11 auto-processing between the enzymatic subunits is required for non-canonical inflammasome signalling

We next sought to determine the caspase-11 cleavage site(s) that generate p32, and whether caspase-11 protease activity mediated cleavage at these sites. E. coli-expressed recombinant caspase-11 auto-cleaves at several sites to generate a variety of proteolytic fragments, and D80 and D285 were previously proposed as candidate auto-processing sites within the CDL or IDL, although this was not verified experimentally (Wang et al, 1996); and indeed an E. coli protease may also process a site within the CARD domain (Ramirez et al, 2018). We identified two further IDL sites, E266 and D277, as potential alternative sites for self-cleavage (Fig 1A). We retrovirally reconstituted caspase-11 expression in Casp11-deficient BMMs, using "CDL-uncleavable" (CDL$^{uncl}$, D80A) or "IDL-uncleavable" (IDL$^{uncl}$, E266A/D277A/D285A) mutants bearing alanine mutations of the putative cleavage sites within either the CDL or the IDL (Fig 1A),

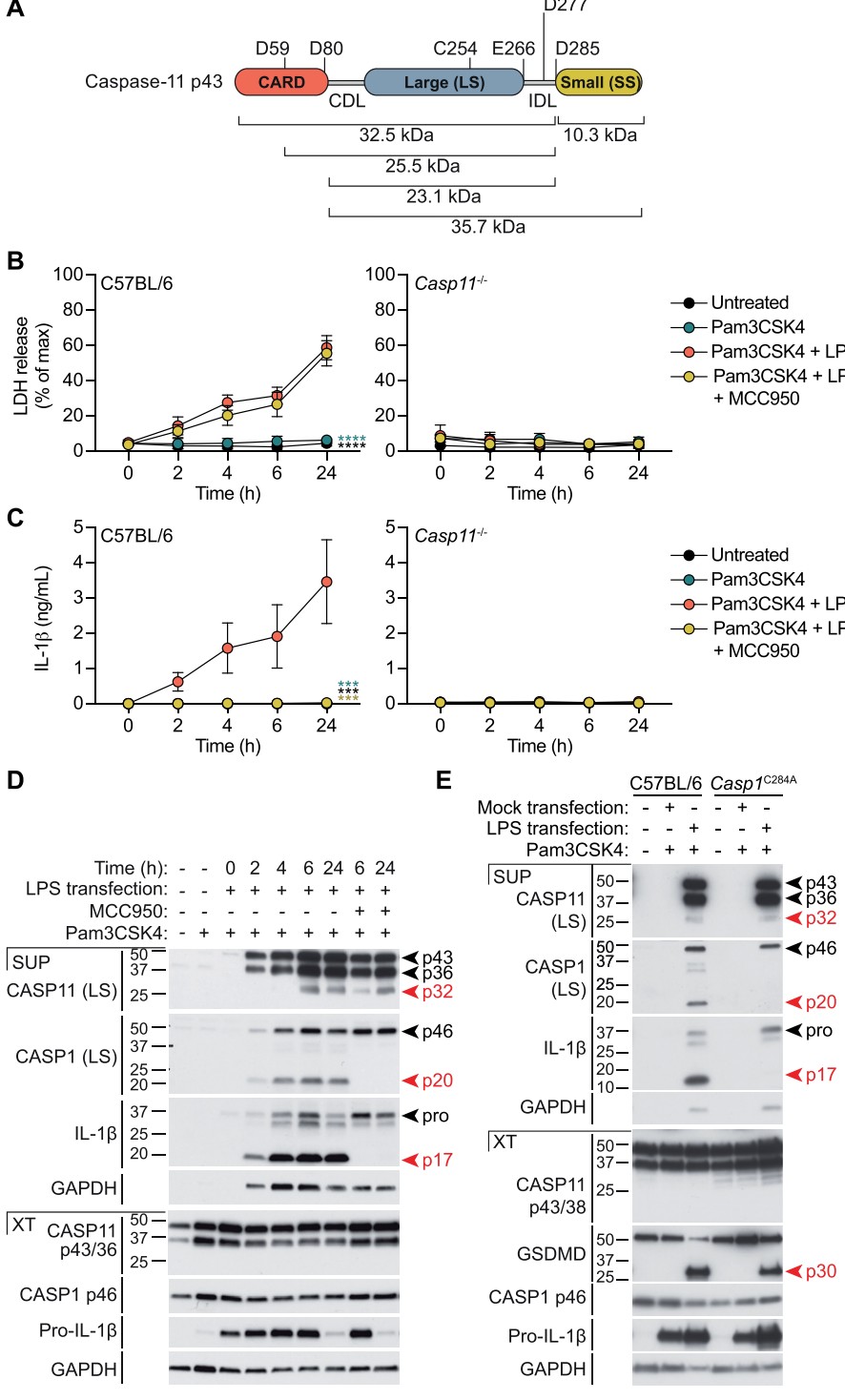

**Figure 1. LPS-induced caspase-11 processing is independent of the NLRP3 and caspase-1 inflammasome.**
**(A)** Domain structure of caspase-11 showing potential caspase cleavage sites, the CDL, IDL and the catalytic cysteine (C254), and the relative predicted molecular weights of caspase-11 fragments. **(B–E)** BMMs were primed for 12 h with Pam$_3$CSK$_4$ (1 $\mu$g/ml) and then transfected with ultrapure K12 *E. coli* LPS (10 $\mu$g/ml) using FuGene HD. MCC950 (10 $\mu$M) was added to cells 30 min before transfection. Supernatants and cell extracts were collected at 8 h post-transfection, or over a time course as indicated. **(B)** Cell death was assessed by quantifying lactate dehydrogenase (LDH) release into the culture medium, compared with a full lysis (Triton X100) control. **(C)** Secretion of mature IL-1$\beta$ into the culture medium was assessed by ELISA. Data in (B–C) are the mean + SEM of three biological replicates, and significance was assessed by two-way ANOVA using the Pam3CSK4+LPS transfection sample as a reference. **(D)** WT BMM or **(E)** WT versus caspase-1 enzyme-dead (C284A) BMM were analysed by immunoblot of the cell culture medium (SUP) and cell extracts (XT). Western blots are representative of three biological replicate experiments.

and compared these with WT caspase-11 and a C254A mutant that renders the protease inactive. Equivalent caspase-11 expression was confirmed in lysates of Pam$_3$CSK$_4$-primed BMM by Western blot (Fig 2A). Pam$_3$CSK$_4$-primed BMMs were transfected with LPS to activate caspase-11, and caspase-11 cleavage was monitored 6 h later. CDL mutation failed to block caspase-11 cleavage to p32 (Fig 2B), suggesting that caspase-11 cleavage at this CDL site is

dispensable for caspase-11 signalling. Instead, LPS transfection failed to induce caspase-11 processing to p32 in BMMs expressing the IDL$^{uncl}$ or enzyme-dead caspase-11 mutants (Fig 2B). This indicates that the p32 cleavage fragment arises from p43 self-cleavage at the IDL. We cannot exclude the possibility that IDL auto-cleavage occurs in tandem with self-processing at D59 to generate a smaller p26 fragment (Wang et al, 1996; Lee et al, 2018),

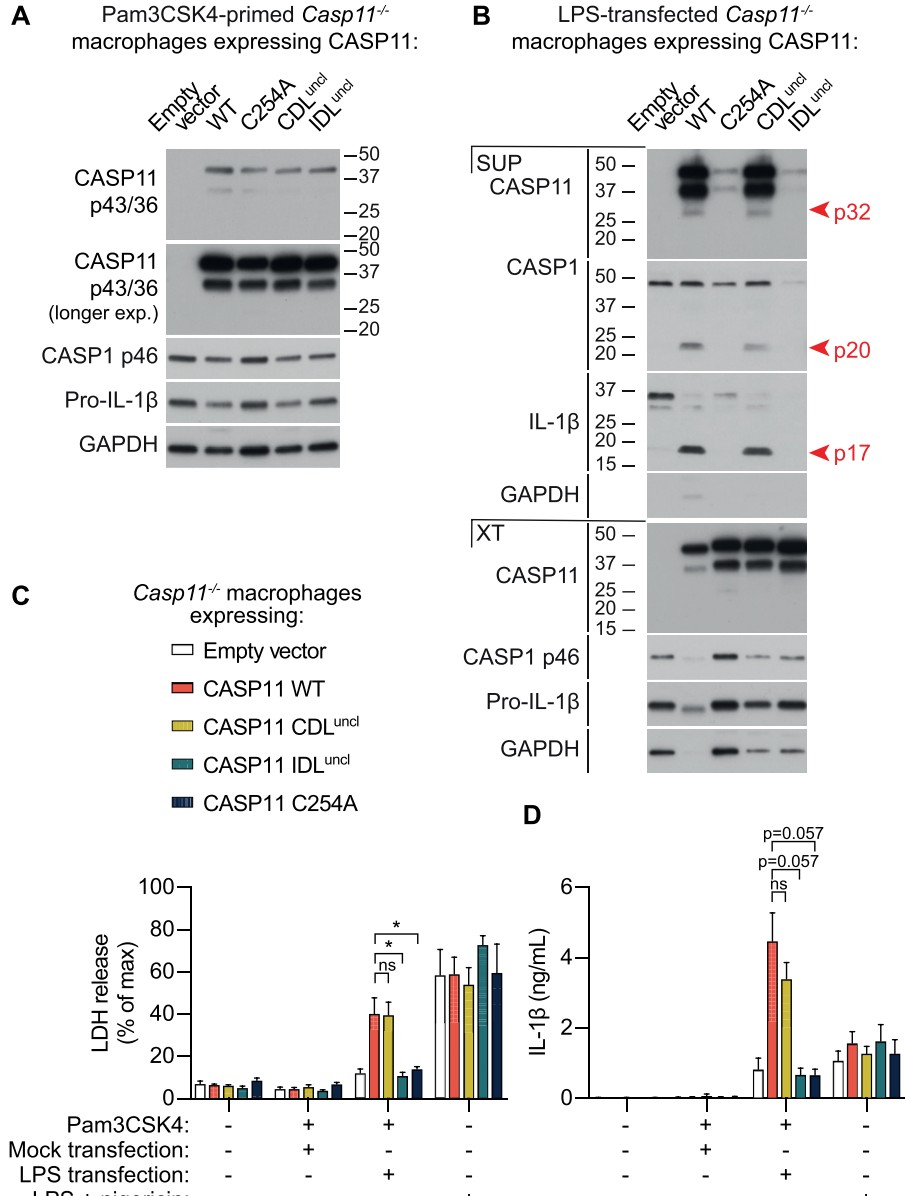

**A** Pam3CSK4-primed *Casp11*⁻/⁻ macrophages expressing CASP11:

**B** LPS-transfected *Casp11*⁻/⁻ macrophages expressing CASP11:

**C** *Casp11*⁻/⁻ macrophages expressing:

□ Empty vector
■ CASP11 WT
■ CASP11 CDL^uncl
■ CASP11 IDL^uncl
■ CASP11 C254A

**D**

**Figure 2. Caspase-11 requires IDL but not CDL processing for inducing cell death and IL-1β release.** Caspase-11 WT, catalytic mutant (C254A), CDL mutant (CDL^uncl), or IDL mutant (IDL^uncl) were retrovirally expressed in *Casp11*⁻/⁻ BMM. Cells were primed for 12 h with Pam₃CSK₄ or 4 h with LPS, and transfected with ultrapure K12 *E. coli* LPS 10 μg/ml for 6 h or exposed to 5 μM nigericin for 2 h. **(A)** Western blot assessed expression of caspase-11 mutants in cell extracts of Pam₃CSK₄-primed, untransfected BMM. **(B)** Immunoblot detected mature IL-1β and the caspase-11 or caspase-1 large subunits in the culture medium (SUP) and cell extracts (XT) of Pam₃CSK₄-primed BMMs transfected with LPS for 6 h. **(C)** Cell death and **(D)** IL-1β secretion was assessed 6 h after LPS transfection or 2 h after nigericin exposure. Western blots are representative of three biological replicate experiments. Graphs are mean + SEM of four biological replicate experiments, with significance assessed using a Mann–Whitney test.

but we feel this is unlikely, as D59 is buried within the CARD domain. To determine the impact of caspase-11 linker cleavage on protease activity, caspase-11-expressing BMM were transfected with LPS and monitored for non-canonical signalling outputs. Cells were also exposed to the canonical NLRP3 agonist, nigericin, where they showed equivalent caspase-11–independent responses, as expected (Fig 2C and D). CDL mutation did not affect the capacity of LPS-activated caspase-11 to drive cell death, or NLRP3-dependent caspase-1 and IL-1β cleavage or IL-1β secretion (Fig 2B–D). By contrast, IDL mutation blocked caspase-11–induced cell death, secretion of mature IL-1β and LPS-induced caspase-1 cleavage (Fig 2B–D). As IDL cleavage was necessary for p32 generation and noncanonical inflammasome signalling, this suggests that caspase-11 requires auto-cleavage at the IDL to generate a fully active p32/p10 species (CARD-LS/SS) that cleaves GSDMD to drive downstream cell death and NLRP3 inflammasome activation.

## Dimerization activates caspase-11

Caspase-1 requires proximity-induced dimerization upon canonical inflammasomes for catalytic activity, and indeed, dissociation from this platform inactivates caspase-1 (Boucher et al, 2018). Dimerization is similarly critical for the acquisition of basal proteolytic activity for the apoptotic initiator caspases, caspase-8 and -9 (Boatright et al, 2003). As cellular caspase-11 requires interaction with LPS via its CARD domain for protease activation (Shi et al, 2014), we hypothesised that LPS binding enables caspase-11 clustering, leading to dimerization of the catalytic subunits and the acquisition of basal proteolytic activity, such as the ability to auto-cleave at the IDL. To test this hypothesis, we used the DmrB system to precisely control homodimerization of caspase-11 enzymatic domains, independently of their interactions with LPS. ΔCARD_Caspase-11 was N-terminally fused to the DmrB domain (Fig 3A), and dimerization

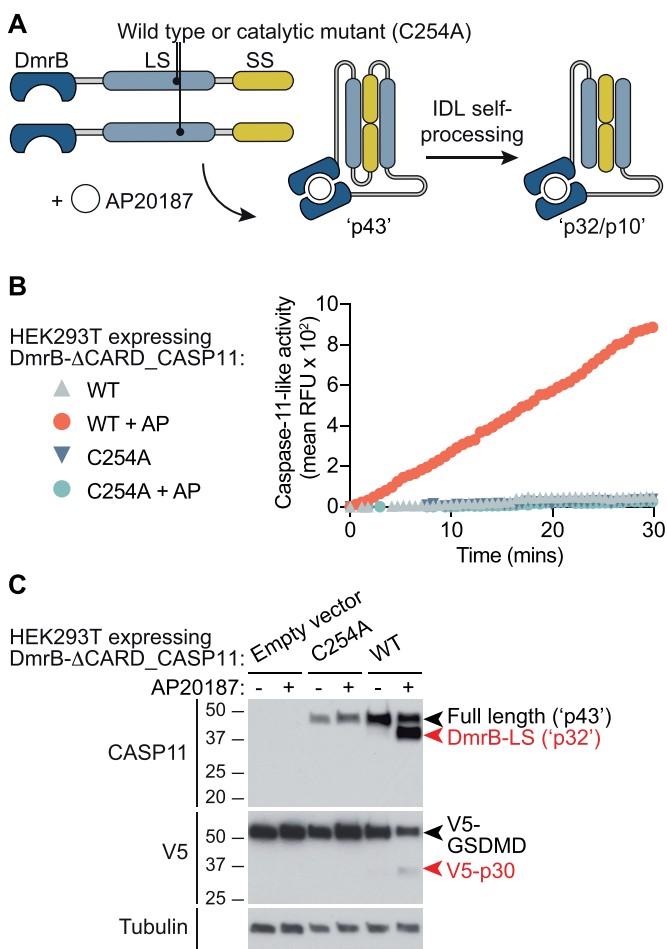

**Figure 3.  Caspase-11 dimerization is necessary and sufficient for auto-cleavage to the p32/p10 species, and protease activity.**
HEK293T cells were transiently transfected with pEF6-DmrB-Caspase-11 constructs depicted in **(A)** to mimic the p43 dimers and p32/p10 species of caspase-11 (actual predicted molecular weights of DmrB fusions are 46 kD and 35/10 kD, respectively). Transfected HEK293T cells were pre-incubated with the dimerizer drug, AP20187, for 30 min before substrate addition. **(B)** Cleavage of AcLEHD-afc by DmrB-Caspase-11 in digitonin-lysed HEK293T cells was monitored over 30 min. Data are mean of technical quadruplicates, and are representative of at least three biological replicate experiments. **(C)** HEK293T cells expressing DmrB-Caspase-11 were incubated with AP20187 for 30 min to induce dimerization. Cells were lysed with digitonin, and incubated with lysates from HEK293T expressing V5-GSDMD for 1 h at 37°C. Samples were precipitated and analysed using an immunoblot. Data are representative of three biological replicate experiments.

was induced by AP20187 to mimic CARD-driven clustering by LPS. DmrB-caspase-11 constructs were expressed in HEK293T cells and caspase-11 activity was measured on peptidic (AcLEHD-afc) and native (V5-GSDMD) substrates. AP20187-induced dimerization of WT, but not C254A, DmrB-caspase-11 triggered caspase-11 auto-cleavage, LEHD-ase activity, and processing of GSDMD to GSDMD-p30 (Fig 3B and C). Dimerization thus induces caspase-11 basal activity, triggering AcLEHD-afc cleavage and self-processing at the IDL. This suggests that LPS induces caspase-11 activity during non-canonical inflammasome signalling by promoting caspase-11 dimerization and IDL auto-processing. It further indicates that while LPS may induce the formation of caspase-11 oligomers, dimers are

both necessary and sufficient for caspase-11 to acquire basal proteolytic activity.

## Caspase-11 auto-processing at D285 is required for substrate cleavage

We next sought to identify the cleavage site(s) within the IDL that generate the p32 fragment during caspase-11 auto-processing. Three single point mutants at candidate sites were thus created within DmrB-caspase-11 (E266A, D277A, and D285A), in addition to the triple mutant (IDL$^{uncl}$) previously shown to abrogate caspase-11 self-cleavage and signalling in macrophages (Fig 2). These constructs were expressed in HEK293T cells, to which AP20187 was added to induce caspase-11 dimerization. The catalytic activity and self-processing of these mutants was then examined. E266A mutation did not affect caspase-11 LEHDase activity, whereas mutation of the catalytic cysteine ablated activity (C254A, Fig 4A and B). Unexpectedly, D277 mutation caused an increase in caspase-11 LEHDase activity (Fig 4A and B), indicating that this mutation may render caspase-11 hyperactive. Mutation of the D285 residue, either as a single mutant (D285A) or within compound mutants (D277A/D285A, IDL$^{uncl}$), markedly diminished but did not ablate caspase-11 LEHDase activity (Fig 4A and B). C254A and D285A mutation suppressed AP20187-induced caspase-11 cleavage (Fig 4B), indicating that D285 is a critical auto-processing site. The D285A mutant was also unable to cleave V5-GSDMD (Fig 4B). Caspase-11 E266A and D277A mutation did not prevent self-processing or GSDMD cleavage (Fig 4B). Together, these data suggest that unprocessed caspase-11 dimers exhibit basal activity, such as the ability to modestly cleave AcLEHD-afc and autoprocess the IDL. Herein, D285 is cleaved first, and is the only cleavage event essential for caspase-11 to process GSDMD. This, however, does not exclude the possibility that E266 and D277 may be cleaved after D285 to further "trim" the IDL, as such small changes to caspase-11 fragment size may not be readily observed by our methods. Such sequential processing has been reported in caspases previously (Boucher et al, 2011). Thus, dimerization is sufficient to induce caspase-11 auto-catalytic activity, but is alone insufficient for inducing the full spectrum of caspase-11 activities, such as GSDMD proteolysis. The latter requires caspase-11 to be both dimeric and auto-cleaved within the IDL, at residue D285.

## The proteolytic activity of caspase-11 dimers unable to self-cleave at the IDL can be rescued by cleavage in *trans*

To confirm the differential requirements for dimerization and IDL cleavage in caspase-11 activities, we generated an engineered form of DmrB-caspase-11 (IDL$^{thr}$) in which two of the candidate cleavage sites (D277A and D285A) were mutated, and a thrombin consensus cleavage site (LVPR/GS) was inserted (Fig 5A). This allowed us to precisely control dimerization and IDL cleavage to p32 separately, with AP20187 and thrombin, respectively. This engineered form of caspase-11 was expressed in HEK293T cells and monitored for self-processing and the capacity to cleavage substrates. As seen with the D285A mutation, caspase-11 IDL mutation (IDL$^{thr}$) blocked self-processing to p32 (Fig 5D). Intriguingly, dimerization of caspase-11

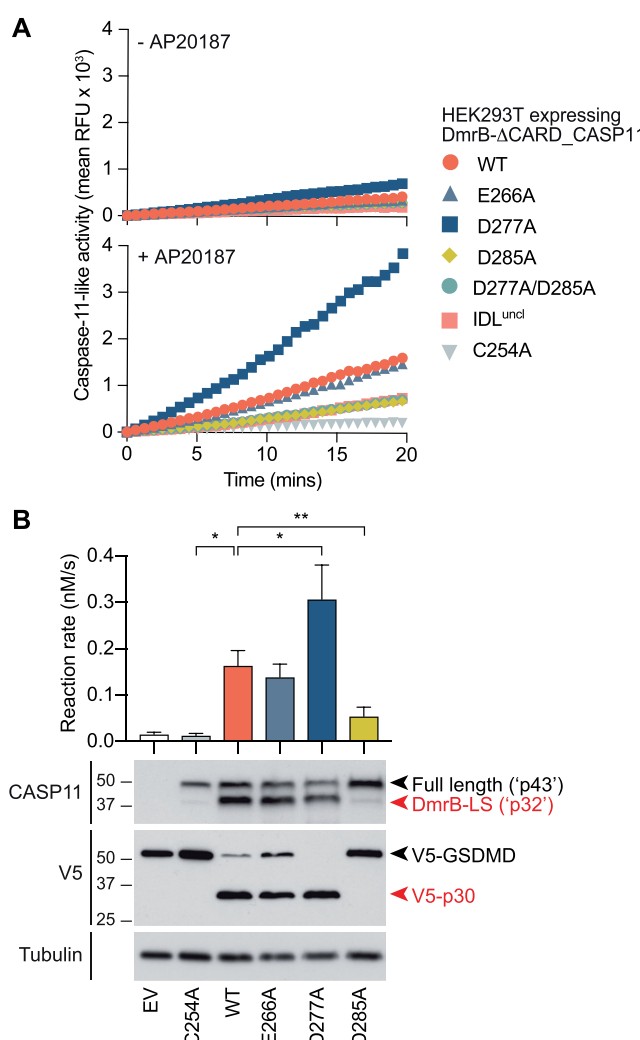

**Figure 4. The MEA/D (D285) cleavage site within the IDL is critical for caspase-11 auto-processing and full protease activity.**
HEK293T cells were transfected with either DmrB alone (empty vector, EV) or DmrB-caspase-11 mutants: WT, C254A (catalytic mutant), IDL[uncl] (IDL triple cleavage mutant; E266A/D277A/D285A), and single IDL mutants: E266A, D277A, and D285A. Cells were exposed to 500 nM AP20187 for 30 min and lysed with digitonin for quantification of proteolytic activity, either by **(A)** kinetics of AcLEHD-afc cleavage over 20 min, or **(B)** reaction rate (upper), and incubation with full-length GSDMD-V5 to assess the extent of cleavage to p30 after 1 h (lower). Data in (B upper) are mean + SEM of four biological replicates. Data were analysed for normality using the Shapiro–Wilk normality test, and tested for significance using parametric paired *t* tests (two-sided). All other data are representative of at least three biological replicates.

IDL[thr] triggered some processing to a minor, shorter fragment (Fig 5D). This is likely a consequence of the hyper-activity of p43 dimers due to D277A mutation, leading to E266 cleavage and generation of a p30/p10 species. This caspase-11 p30/p10 species, however, appeared to be inactive, as it did not cleave AcLEHD-afc (Fig 5C) or GSDMD (Fig 5D). Dimerized IDL[thr] did not induce V5-GSDMD or AcLEHD-afc cleavage unless it was first incubated with thrombin, whereas thrombin did not affect the activities of WT or C254A caspase-11 (Fig 5B–D). Addition of thrombin without AP20187 pre-incubation lead to a modest increase in caspase-11 IDL[thr] LEHDase

activity (Fig 5B), suggesting that IDL cleavage may facilitate caspase-11 proteolytic activity, possibly by promoting the dimerization of caspase-11 monomers. Together, these results support a model whereby caspase-11 dimerization induces basal proteolytic activity such as the ability to self-cleave, whereafter IDL processing is both necessary and sufficient to generate the fully active p32/p10 species of caspase-11 dimer, leading to cleavage of GSDMD and non-canonical inflammasome signalling.

## Discussion

Inflammatory pathways of the innate immune system provide defence against microbial infection. Innate immune cells require mechanisms which rapidly detect and respond to cytosolic bacteria, while limiting indiscriminate collateral damage. Caspase-11 provides important surveillance of the host cytosol in macrophages, dendritic cells, neutrophils and epithelia (Hagar et al, 2013; Kayagaki et al, 2013; Knodler et al, 2014; Oficjalska et al, 2015; Zanoni et al, 2016). Although the signalling events leading up to and following caspase-11 activation are well understood, the precise molecular mechanisms governing caspase-11 activation and substrate repertoire remain unclear. Here we use inducible systems to control caspase-11 dimerization and cleavage, and demonstrate that both dimerization and IDL auto-cleavage at residue D285 are required for caspase-11 to cleave GSDMD and thus drive cell death (pyroptosis, NETosis) during non-canonical inflammasome signalling.

During the preparation of this manuscript, the D285 residue within the caspase-11 IDL was identified as important for caspase-11 function in vivo (Lee et al, 2018). This study supports our conclusions that caspase-11 is cleaved at D285 and that this is critical for non-canonical inflammasome signalling ouputs. Lee et al (2018) did not investigate the function of other candidate self-cleavage sites within the IDL or CDL, or elucidate the mechanism by which caspase-11 acquires basal activity. Lee et al (2018) proposed that active caspase-11 contains a p26 large subunit fragment generated by auto-cleavage of both D59 and D285 sites, or alternatively, by D285 auto-cleavage of p36, a short form of caspase-11 derived from an M61 alternative start site that is lacking most of the LPS-interacting CARD domain. While the precise identity of the caspase-11 cleavage fragment encompassing the LS (p32 versus p26) that is observed in both studies is not resolved in either study, we believe that the active species of caspase-11 is likely to be p32/p10 rather than p26/p10, because: (i) caspase-11 p36 cannot bind LPS (Shi et al, 2014), so p36 caspase-11 would not be expected to dimerize or acquire the capacity to auto-cleave at the IDL to generate p26/p10. By contrast, p43 can bind LPS and so become activated to generate p32/p10; and (ii) the D59 cleavage site proposed by Lee et al (2018) is located within an α helix of the CARD (Fig S1), and so is unlikely to be a target of auto-cleavage, as caspases prefer to cleave in flexible loop regions (Timmer et al, 2009).

Our data give new insight into the molecular events underlying activation of caspase-11 protease activity by intracellular LPS. We propose a model in which LPS interaction allows caspase-11 to cluster, leading to proximity-induced dimerization of the catalytic

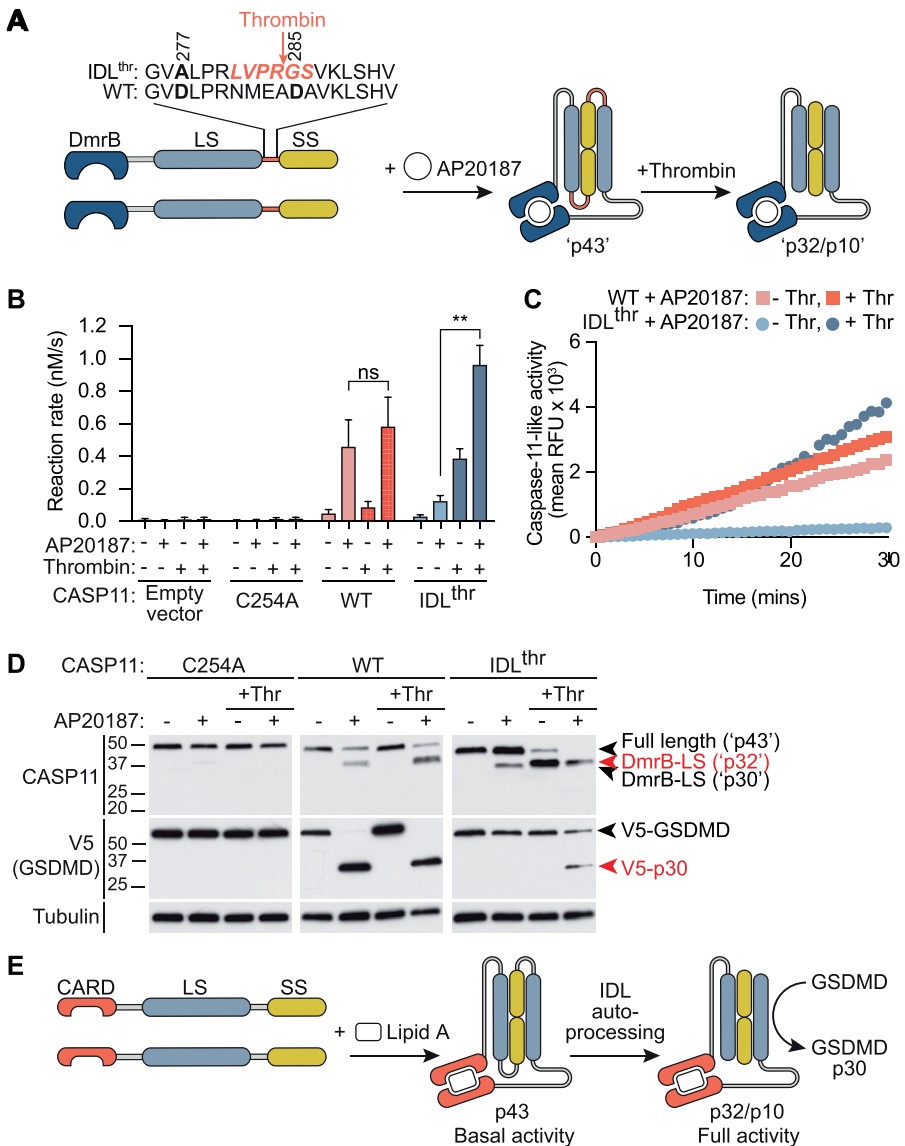

**Figure 5. Cleavage of the IDL in *trans* promotes caspase-11 cleavage of GSDMD and AcLEHD-afc.** HEK293T cells were transiently transfected with constructs containing DmrB-Caspase-11, WT, C254A, and IDL[thr], in which a thrombin cleavage site replaces the caspase-11 IDL auto-processing site, as depicted in **(A)**, to allow generation of unprocessed dimers (analogous to caspase-11 p43; actual predicted weight for the DmrB-caspase-11 fusion, ~46 kD) and IDL-cleaved dimers (analogous to caspase-11 p32/p10, actual predicted weight 35/10 kD). **(B)** Cells were incubated with AP20187 (500 nM) for 30 min, and then AcLEHD-afc cleavage was measured with and without the addition of thrombin (20 U/ml) to the reaction. Data are mean + SEM of four biological replicates. Data were analysed for normality using the Shapiro–Wilk normality test, and tested for significance using parametric paired *t* tests (two-sided). **(C)** AcLEHD-afc kinetic trace of AP20187-treated cells expressing DmrB-Caspase-11 WT versus IDL[thr], in the presence and absence of thrombin (20 U/ml) in the reaction. **(D)** HEK293T expressing the DmrB-Caspase-11 constructs were exposed to AP20187 for 30 min, and then lysed and incubated for 15 min with thrombin (20 U/ml) before the addition of V5-GSDMD for 1 h. **(E)** Model for LPS-induced caspase-11 dimerization, auto-processing, and activation.

subunits. This dimer of full-length caspase-11 (p43) has basal activity (for example, it can self-cleave) but a restricted substrate repertoire, as it cannot process GSDMD. Dimerization-induced auto-proteolysis at the IDL generates the fully active species of caspase-11 dimer (which we propose to be p32/p10) that can cleave GSDMD to initiate cell death (Fig 5E). It is likely that caspase-11 p32/p10 also cleaves additional substrates with important functions in cell death and host defence. Caspase-4 is similarly cleaved to generate a p32 fragment upon exposure to cytosolic LPS or Gram-negative bacteria (Casson et al, 2015), suggesting the caspase-11 signalling mechanism we describe here is also conserved during non-canonical inflammasome signalling in humans.

This proposed mechanism for caspase-11 activation has parallels to the mechanism by which caspase-1 is activated upon canonical inflammasomes. Caspase-1 is recruited to canonical inflammasomes via CARD–CARD interactions and caspase-1 clustering upon this complex leads to proximity-induced dimerization

(Boucher et al, 2018). The uncleaved (p46) species of caspase-1 can initiate GSDMD-dependent cell death (Broz et al, 2010). By contrast, caspase-11 requires both dimerization and IDL processing to generate the fully active caspase-11 species able to cleave GSDMD and drive cell death. The differing substrate repertoires of the p43 versus p32/p10 species of caspase-11 is reminiscent of both caspases-1 and -8, where signalling outcomes are altered if IDL cleavage fails to occur (Broz et al, 2010; Oberst et al, 2010). For instance, caspase-1 p46 dimers induce cell death without maturation of IL-1β and IL-18 (Broz et al, 2010). The future identification of caspase-11 substrates that can be processed by "uncleaved," yet dimerized caspase-11 will be of great interest. Our data using a small peptidic caspase-11 substrate (AcLEHD-afc) indicate that caspase-11 p43 dimers may be intrinsically less catalytically active and therefore less able to cleave "suboptimal" cleavage sites within potential substrates. It is also possible that caspase-11 p43 dimers are relatively unstable, as IDL cleavage of other caspases stabilizes

their active sites and substrate binding pockets (Fuentes-Prior & Salvesen, 2004; Boucher et al, 2011). Alternatively, IDL cleavage may expose important recognition sites for interaction with particular protein substrates, to alter substrate specificity. The impact of caspase-11 auto-processing on substrate repertoire may be particularly important when considering that some ligands of caspase-11, such as oxPAPC, do not induce caspase-11 cleavage or pyroptosis (Zanoni et al, 2016).

Over-expressed caspase-11 is reported to cleave at the CDL (Kayagaki et al, 2011; Wang et al, 1996). We expressed caspase-11 in macrophages or HEK293T at near-physiological concentrations, and did not find evidence for CDL auto-cleavage at any site, including the candidate D80 residue between the CARD domain and the linker sequence to the protease domain. Importantly, D80 mutation to prevent processing at this site did not suppress macrophage non-canonical inflammasome signalling. A structural model for the caspase-11 CARD domain suggests that the D80 putative cleavage site is located within the terminal region of the final $\alpha$ helix, abutting the CDL (Fig S1). D59 was recently proposed to be sensitive to self-proteolysis, but this residue is located within the CARD domain, and so is also likely to be inaccessible for auto-cleavage (Fig S1). It is thus unlikely that caspase-11 auto-processes at D59 or D80 under physiological conditions. By contrast, caspase-1 does auto-cleave at the CDL, leading to the release of caspase-1 dimers from the inflammasome and terminating protease activity. Given that caspase-11 signalling is not limited by CDL auto-cleavage, this suggests that caspase-11 activity may be physically confined to the LPS-complex, which may restrict substrate availability, as substrates would require recruitment to the non-canonical inflammasome for processing. This also raises the question of whether caspase-11 activity may be modulated by CDL cleavage in *trans*, by other proteases. Cathepsin G and granzyme B both cleave caspase-11, although the precise cleavage sites have not been identified (Wang et al, 1996; Schotte et al, 1998; Chen et al, 2018b). If these proteases target the CDL they could mediate caspase-11 inactivation, by analogy to caspase-1 deactivation mechanisms (Boucher et al, 2018). Alternatively, if these proteases target the IDL, they may support caspase-11 activity. For example, caspase-8 processing by cathepsin D promotes caspase-8 dimerization and subsequent activity in neutrophils (Conus et al, 2012).

In summary, caspase-11 processing is often monitored as a proxy for caspase-11 activation. Here, we confirm that caspase-11 is autoprocessed at D285 to generate the fully active protease species. Our data indicate that caspase-11 gains activity within the non-canonical inflammasome via a two-step mechanism involving first dimerization and then IDL auto-processing. Such a mechanism ensures controlled and appropriate caspase-11 activation during cytosolic Gram-negative bacterial infection.

# Materials and Methods

### Mice

All mice were housed in specific pathogen-free facilities at the University of Queensland. $Casp11^{-/-}$ (Kang et al, 2002) mice were backcrossed at least 10 times to C57BL/6. The $Casp1^{C284A/C284A}$ line, in which the catalytic cysteine is mutated to render caspase-1 enzymatically inactive, was generated via CRISPR/Cas9 gene editing of C57BL/6 mice at the University of Queensland Facility for Advanced Genome Editing. Mice were used as a source of primary bone marrow progenitors. The University of Queensland's Animal Ethics Committee approved all experimental protocols.

### Murine macrophage inflammasome assays

WT (C57BL/6), $Casp11^{-/-}$ or $Casp1^{C284A}$ murine BMMs were differentiated from bone marrow progenitors as previously described (Schroder et al, 2012). BMM were plated at a density of $1 \times 10^6$ cells/ml in complete macrophage media (RPMI-1640, 10% fetal bovine serum, 1× Glutamax, and 150 ng/ml endotoxin-free recombinant CSF-1), and were primed for 12 h with 1 $\mu$g/ml Pam$_3$CSK$_4$. The medium was then replaced with Opti-MEM replete with 150 ng/ml CSF-1, before cells were transfected with 10 $\mu$g/ml ultrapure K12 *E. coli* LPS (0.25% FuGENE HD Transfection Reagent; Promega) for the indicated times. To activate the NLRP3 inflammasome, BMM were first primed for 4 h with 100 ng/ml K12 ultrapure LPS, before the medium was replaced with CSF-1-replete Opti-MEM containing 5 $\mu$M Nigericin Sodium salt (Sigma-Aldrich). IL-1$\beta$ secretion into the cell culture medium was assessed by ELISA (eBioscience IL-1$\beta$ Ready-SET-Go!), according to manufacturer's instructions. Cell cytotoxicity was measured using the CytoTox96 Non-radioactive Cytotoxicity Assay (Promega) and expressed as a percentage of total cellular LDH (100% lysis control). Cell extracts and methanol/chloroform-precipitated supernatants were analysed by Western blot using standard methods (Gross, 2011), using antibodies against the caspase-11 large subunit (EPR18628, 1:1,000; Abcam), mIL-1$\beta$ (polyclonal goat antibody, 1:1,000; R&D Systems), caspase-1 large subunit (casper-1, 1:1,000; Adipogen), V5 (SV5-Pk1, 1:2,000; AbD Serotec), $\alpha$-tubulin (B5-1-2, 1:2,000; Sigma-Aldrich), and GAPDH (polyclonal rabbit antibody, 1:5,000; BioScientific).

### Retroviral transduction

The coding sequence of caspase-11 was cloned into a replication defective mouse stem cell construct (pMSCV). Caspase-11 mutants were generated by PCR mutagenesis. The PlatE cell line was used to produce and package the retrovirus. PlatE cells were transfected with pMSCV vectors using Lipofectamine 2000, and incubated for 48 h at 32°C and 5% CO$_2$ for virus production. PlatE supernatants were filtered (0.45 $\mu$m), supplemented with 6 $\mu$g/ml polybrene, 20 mM Hepes and 150 ng/ml CSF1, and used to spin-infect $Casp11^{-/-}$ bone marrow progenitors on day 2 of their CSF-1-directed differentiation.

### HEK-293T transfection and caspase-11 dimerization

The DmrB-ΔCARD-caspase-11 mutants were cloned into the pEF6 vector. HEK293T cells (ATCC CRL-3216) were transfected with these constructs using lipofectamine, and cells were reseeded at $1 \times 10^6$ cells/ml. Transfected cells were then incubated in opti-MEM containing the 500 nM of the B/B Homodimerizer (AP20187; Clontech) for 30 min. The medium was then replaced with caspase

activity buffer (200 mM NaCl, 50 mM Hepes pH 8.0, 50 mM KCl, 100 μg/ml digitonin, 10 mM DTT) supplemented with 100 μM AcLEHD-afc or V5-GSDMD-expressing HEK293T cell extract. For experiments in which caspase-11 was cleaved with thrombin, 20 U/ml bovine thrombin (Sigma-Aldrich) was added to the caspase activity buffer 15 min before V5-GSDMD addition, or the same time as AcLEHD-afc addition. Hydrolysis of the caspase-11 substrate AcLEHD-afc was monitored at 37°C at regular time intervals using the M1000 TECAN spectrofluorometer (400 nm excitation, 505 nm emission). V5-GSDMD cleavage was monitored after 2 h or as indicated. Cell extracts and supernatants were precipitated using methanol/chloroform and analysed by immunoblotting using standard procedures (Gross, 2011).

### Data analysis and statistics

Statistical analysis was performed using GraphPad Prism 6.0 software. Data were analysed for normality using the Shapiro–Wilk normality test, and tested for statistical significance using parametric paired $t$ tests (two-sided), nonparametric Mann–Whitney tests, or two-way ANOVA (for time course analysis). LEHDase activity curves were analysed by linear regression on the linear portion of the kinetic traces, to determine the slope (relative fluorescent units/second). The relative fluorescent units/second was then converted to reaction rate (nM/s) using an AFC standard curve.

# Supplementary Information

# Acknowledgments

This work was supported by a Project Grant (DP160102702) from the Australian Research Council to K Schroder. D Boucher was supported by a postdoctoral fellowship from the University of Queensland. AH Chan was supported by a postgraduate research training program scholarship from the Australian Government. K Schroder was supported by an Australian Research Council Future Fellowship (FT130100361) and a National Health and Medical Research Council of Australia Fellowship (1141131).

## Author Contributions

C Ross: conceptualization, data curation, formal analysis, investigation, and writing—original draft, review, and editing.
AH Chan: data curation, formal analysis, investigation, methodology, and writing—review and editing.
J Von Pein: data curation, formal analysis, investigation, and writing—review and editing.
D Boucher: conceptualization, data curation, formal analysis, supervision, investigation, methodology, and writing—review and editing.
K Schroder: conceptualization, resources, data curation, formal analysis, supervision, funding acquisition, project administration, and writing—original draft.

## Conflict of Interest Statement

K Schroder is a co-inventor on patent applications for NLRP3 inhibitors which have been licensed to Inflazome Ltd, a company headquartered in Dublin, Ireland. Inflazome is developing drugs that target the NLRP3 inflammasome to address unmet clinical needs in inflammatory disease. K Schroder served on the Scientific Advisory Board of Inflazome in 2016–2017. The authors have no further conflicts of interest to declare.

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
