## [Reviewer comments · Life Science Alliance]

Dimerization and auto-processing induce caspase-11 protease activation within the non-canonical inflammasome

Connie Ross, Amy H. Chan, Jessica Von Pein, Dave Boucher and Kate Schroder

DOI: 10.26508/lisa.00237

Review timeline:

Submission Date:	11 November 2018
Editorial Decision:	12 November 2018
Revision Received:	28 November 2018
Editorial Decision:	29 November 2018
Accepted:	30 November 2018

Report:

(Note: Letters and reports are not edited. The original formatting of letters and referee reports may not be reflected in this compilation.)

Please note that the manuscript was previously reviewed at another journal and the reports were taken into account in inviting a revision for publication at *Life Science Alliance* prior to submission to *Life Science Alliance*.

No Peer Review Process File is available with this article, as the authors have chosen not to make the review process public in this case.

1st Editorial Decision

12 November 2018

Thank you for transferring your manuscript entitled "Dimerization and auto-processing induce caspase-11 activation within the non-canonical inflammasome" to Life Science Alliance. The manuscript was assessed by expert reviewers at another journal before, and those review reports have been transferred to us with your permission.

The reviewers who assessed your work at the other journal were concerned by a similar study recently published by others. They appreciated, however, that your work provides additional insight. Based on the reviewer reports already at hand, we would like to invite you to submit a revised version for publication in Life Science Alliance. Additional experimental work as suggested by the reviewers is not needed. We would, however, expect a point-by-point response to all concerns raised and accordingly changes to the manuscript text (especially to clarify the fragment sizes observed (reviewer #2)) and the data representation (eg less cropped western blots).

Thank you for this interesting contribution to Life Science Alliance. We are looking forward to receiving your revised manuscript.

2nd Editorial Decision

29 November 2018

Thank you for submitting your revised manuscript entitled "Dimerization and auto-processing induce caspase-11 activation within the non-canonical inflammasome". I appreciate the way you responded to the concerns previously raised by the reviewers and the changes made to the manuscript.
